# Phillygenin Suppresses Glutamate Exocytosis in Rat Cerebrocortical Nerve Terminals (Synaptosomes) through the Inhibition of Ca_v_2.2 Calcium Channels

**DOI:** 10.3390/biomedicines12030495

**Published:** 2024-02-22

**Authors:** Ming-Yi Lee, Tzu-Yu Lin, Ya-Ying Chang, Kuan-Ming Chiu, Su-Jane Wang

**Affiliations:** 1Department of Medical Research, Far-Eastern Memorial Hospital, New Taipei City 22060, Taiwan; mingyi.lee@gmail.com; 2Department of Anesthesiology, Far-Eastern Memorial Hospital, New Taipei City 22060, Taiwan; drlin1971@gmail.com (T.-Y.L.); yychang0310@saturn.yzu.edu.tw (Y.-Y.C.); 3Department of Mechanical Engineering, Yuan Ze University, Taoyuan 32003, Taiwan; 4International Program in Engineering for Bachelor, Yuan Ze University, Taoyuan 32003, Taiwan; 5Division of Cardiovascular Surgery, Cardiovascular Center, Far-Eastern Memorial Hospital, New Taipei City 22060, Taiwan; 6Department of Electrical Engineering, Yuan Ze University, Taoyuan 32003, Taiwan; 7School of Medicine, Fu Jen Catholic University, New Taipei City 24205, Taiwan; 8Research Center for Chinese Herbal Medicine, College of Human Ecology, Chang Gung University of Science and Technology, Taoyuan 33303, Taiwan

**Keywords:** phillygenin, glutamate exocytosis, voltage-dependent Ca^2+^ channel, cerebral cortex, synaptosomes

## Abstract

Glutamate is a major excitatory neurotransmitter that mediates neuronal damage in acute and chronic brain disorders. The effect and mechanism of phillygenin, a natural compound with neuroprotective potential, on glutamate release in isolated nerve terminals (synaptosomes) prepared from the rat cerebral cortex were examined. In this study, 4-aminopyridine (4-AP), a potassium channel blocker, was utilized to induce the release of glutamate, which was subsequently quantified via a fluorometric assay. Our findings revealed that phillygenin reduced 4-AP-induced glutamate release, and this inhibitory effect was reversed by removing extracellular Ca^2+^ or inhibiting vesicular transport with bafilomycin A1. However, exposure to the glutamate transporter inhibitor dl-threo-beta-benzyl-oxyaspartate (dl-TOBA) did not influence the inhibitory effect. Moreover, phillygenin did not change the synaptosomal membrane potential but lowered the 4-AP-triggered increase in intrasynaptosomal Ca^2+^ concentration ([Ca^2+^]_i_). Antagonizing Ca_v_2.2 (N-type) calcium channels blocked the inhibition of glutamate release by phillygenin, whereas pretreatment with the mitochondrial Na^+^/Ca^2+^ exchanger inhibitor, CGP37157 or the ryanodine receptor inhibitor, dantrolene, both of which block intracellular Ca^2+^ release, had no effect. The effect of phillygenin on glutamate release triggered by 4-AP was completely abolished when MAPK/ERK inhibitors were applied. Furthermore, phillygenin attenuated the phosphorylation of ERK1/2 and its major presynaptic target, synapsin I, a protein associated with synaptic vesicles. These data collectively suggest that phillygenin mediates the inhibition of evoked glutamate release from synaptosomes primarily by reducing the influx of Ca^2+^ through Ca_v_2.2 calcium channels, thereby subsequently suppressing the MAPK/ERK/synapsin I signaling cascade.

## 1. Introduction

Glutamate is a major excitatory neurotransmitter of the central nervous system and the most abundant neurotransmitter in the brain. Glutamate plays a crucial role in synaptic plasticity, learning, and memory [1,2]. Maintaining optimal glutamate levels is essential, as excitotoxicity caused by excessive glutamate release induces an increase in intracellular Ca^2+^ levels. This event, in turn, initiates a cascade of reactions inside the cell, including increased oxygen free radical formation, impaired mitochondrial function, and protease activation, ultimately leading to cell death [3,4]. Pathology can occur in numerous neurological disorders, such as ischemia, traumatic brain injury, epileptic seizures, Alzheimer’s disease, Parkinson’s disease, and amyotrophic lateral sclerosis [4,5,6,7]. Therefore, reducing the release of glutamate from nerve terminals is a promising strategy to protect against neurological disorders linked to excitotoxicity-related pathologies.

An increasing number of studies suggest that medicinal plants are attractive sources of molecules for the development of novel pharmaceuticals and have shown promising results in the prevention and treatment of brain disorders [8,9]. Phillygenin is a lignan compound extracted from the medical herb *Forsythia suspensa* that is traditionally used to treat inflammation, pain, fever, nausea, vomiting, and abscesses [10]. The pharmacokinetics of phillygenin in rats exhibit first-order kinetics, with rapid distribution and elimination, while in mice, also shows high oral bioavailability, peaking within 30 min [11,12]. Previous studies have shown that phillygenin has diverse biological activities, including anti-inflammatory, antioxidant, antitumor, antibacterial, antiviral, analgesic, and hepatoprotective effects [13,14,15,16,17,18,19,20]. Since free radical-induced oxidative damage to the brain is recognized as a primary cause of neuronal death in various neurodegenerative disorders, compounds with antioxidative properties, such as phillygenin, may be effective at preventing or delaying these central nervous system (CNS) disorders [21,22]. Moreover, phillygenin possesses anti-inflammatory properties [23], which may provide neuroprotective benefits by potentially reversing cellular damage and slowing the progression of neuronal cell loss in individuals with neurodegenerative disorders [24]. Therefore, we predicted that phillygenin may protect against glutamate-induced neuronal excitotoxicity.

Since the excessive release of glutamate constitutes a pivotal factor in the pathogenesis of neurological diseases, in this study, we aimed to investigate the impact of phillygenin on glutamate release. Isolated rat cerebral cortex nerve terminals (synaptosomes), which is a well-established model for studying synaptic transmission, were utilized in this study. In particular, synaptosome preparations can accumulate, store, and release neurotransmitters without any postsynaptic interactions. Using this model, we further explored the synaptosomal plasma membrane potential, activation of voltage-dependent Ca^2+^ channels (VDCCs), the intrasynaptosomal Ca^2+^ concentration ([Ca^2+^]_i_), and the potential underlying mechanisms of phillygenin on evoked glutamate release.

## 2. Materials and Methods

### 2.1. Chemicals and Reagents

All chemicals and reagents used in this work were of analytical grade. Sucrose, glucose, L-glutamate, 4-aminopyridine (4-AP), ethylene-bis(oxyethylenenitrilo)tetraacetic acid (EGTA), sodium dodecyl sulfate (SDS), β-nicotinamide adenine dinucleotide phosphate disodium salt (NADP), glutamate dehydrogenase (GDH), and dimethyl sulfoxide (DMSO) were purchased from Sigma–Aldrich (St. Louis, MO, USA). 3,3′-Dipropylthiadicarbocyanine iodide (DiSC3(5)), fura-2 acetoxymethyl ester (Fura-2 AM), Halt™ protease and phosphatase inhibitor cocktail, and the Pierce™ BCA protein assay kit were obtained from Thermo Fisher Scientific (Waltham, MA, USA). Bafilomycin A1, dl-threo-β-benzyloxyaspartate (dl-TBOA), dantrolene, 7-chloro-5-(2-chlorophenyl)-1,5-dihydro-4,1-benzothiazepin-2(3H)-one (CGP37157), N-(2-(4-bromocinnamylamino)ethyl)-5-isoquinolinesulfonamide (H89), 2-(2-amino-3-methoxyphenyl)-4H-1-benzopyran-4-one (PD98059), 3-[1-[3-(dimethylamino)propyl]indol-3-yl]-4-(1H-indol-3-yl)pyrrole-2,5-dione (GF109203X), ω-conotoxin GVIA, and ω-agatoxin IVA were obtained from Tocris Cookson (Bristol, UK). Phillygenin (≥98%) was obtained from ChemFaces (Wuhan, Hubei, China).

### 2.2. Animals and Ethics

All animal work was performed in accordance with the Guide for the Care and Use of Laboratory Animals of the National Research Council (8th edition, 2011) and approved by the Animal Care and Utilization Committee of Far Eastern Memorial Hospital (approval numbers IACUC-2022-FEMH-03 and IACUC-2023-FEMH-02). Efforts were made to minimize animal pain and distress and reduce the number of animals utilized. Adult male Sprague–Dawley rats (150–200 g) were used in these studies (BioLASCO Taiwan Co., Ltd., Taipei, Taiwan). All rats were maintained in environmentally controlled rooms (22 ± 1 °C; 50% humidity) with diurnal lighting on a 12 h light/dark cycle and with free access to fresh, clean drinking water and food.

### 2.3. Synaptosome Isolation

The synaptosomes utilized in this study were purified from the rat cerebral cortex through the discontinued Percoll gradient procedure [25,26]. Briefly, animals were sacrificed by decapitation, after which the brain was removed and placed on a chilled Petri dish. The cerebral cortex was then dissected and placed in cold isotonic sucrose homogenization buffer (0.32 M sucrose and 4 mM HEPES-NaOH, pH 7.5) and homogenized with a Potter–Elvehjem tissue homogenizer (capacity: 55 mL). The following procedures were carried out at a temperature of 4 °C. The homogenate was centrifuged (3000× *g*, 10 min) to remove debris. Following centrifugation at 15,000× *g* for 10 min, the crude synaptosomal pellet was resuspended in ice-cold sucrose buffer. The resulting supernatant was gently layered on top of a discontinuous Percoll gradient, consisting of layers with sucrose concentrations of 3%, 10%, and 23%, and then centrifuged at 33,000× *g* for 7 min. The pure synaptosomal fraction was obtained from the interface of 10% sucrose and 23% sucrose. After washing with 30 mL of HEPES-buffered media (HBM, containing 20 mM HEPES, 140 mM NaCl, 5 mM NaHCO_3_, 1.2 mM Na_2_HPO_4_, 5 mM KCl, 1 mM MgCl_2_, and 10 mM glucose; pH 7.4), the synaptosomes were further centrifuged at 27,000× *g* for 10 min to remove Percoll. The synaptosomal pellets were then resuspended in HBM buffer, and the protein concentration was determined using a Pierce™ BCA protein assay kit (Thermo Fisher Scientific, Waltham, MA, USA).

### 2.4. Measurement of Glutamate Release

A continuous fluorometric assay was used to assess glutamate release [26,27]. The synaptosomal pellets (0.5 mg) were reconstituted in 2 mL of HBM buffer containing 16 mM bovine serum albumin (BSA). The resulting mixture was placed in a temperature-controlled cuvette with continuous stirring at a constant temperature of 37 °C. After 5 min of stirring, 2 mM NADP, 50 U of GDH, and either 1 mM CaCl_2_ or 0.3 mM EGTA were added. After an additional 5 min of incubation, 1 mM 4-AP or 15 mM KCl was added to stimulate glutamate release. The oxidative decarboxylation of the released glutamate, which resulted in NADP reduction, was monitored by measuring NADPH fluorescence at excitation and emission wavelengths of 340 and 460 nm, respectively. The data were collected at 2-s intervals. A standard of exogenous glutamate (5 nmol) was added at the conclusion of the experiment, and the change in fluorescence was used to calculate the amount of released glutamate; this value was expressed as nmol mg^−1^ protein. Glutamate release was calculated until the fluorescence reached an equilibrium (approximately 5 min). Cumulative data were analyzed using GraphPad Prism (version 8.4.3; Boston, MA, USA).

### 2.5. Measurement of Synaptosomal Plasma Membrane Potential

DiSC3(5), a carbocyanine dye that responds to voltage changes, was used to measure the electric potential of the nerve ending membrane [28]. This positively charged dye gathers on hyperpolarized membranes and relocates into the lipid bilayer of the synaptosomal plasma membrane. Upon membrane depolarization, DiSC3(5) released from the membrane bilayer results in a rapid increase in green fluorescence [29]. Synaptosomes were resuspended in 2 mL of HBM and incubated in a stirred thermostatted cuvette at 37 °C in a Perkin-Elmer FL-6500 fluorescence spectrophotometer (PerkinElmer, Inc., Waltham, MA, USA). After 5 min, 5 μM DiSC3(5) was added to allow for maximal dye uptake. Following an additional 3 min of incubation, 1.2 mM CaCl_2_ was introduced into the cuvette before initiating depolarization with 1 mM 4-AP for 10 min. DiSC3(5) fluorescence was monitored using a FL-6500 spectrofluorometer with an excitation wavelength of 646 nm and an emission wavelength of 674 nm. The results are presented as arbitrary fluorescence units, with each accumulation period set at 2-s intervals; the data were analyzed using GraphPad Prism (version 8.4.3; Boston, MA, USA).

### 2.6. Measurement of Intrasynaptosomal Ca^2+^ Concentration ([Ca^2+^]_i_)

Fura 2-AM, which is a calcium chelator and fluorescent probe, was used to monitor dynamic changes in cytosolic free calcium in synaptosomes [27]. In brief, synaptosomal pellets were reconstituted in 2 mL of HBM. Subsequently, 5 μM Fura-2-AM, 100 μM CaCl_2_, and 16 μM BSA were added. Following a 30-min incubation at 37 °C, the synaptosomes were centrifuged for 1 min at 10,000× *g* to eliminate excess Fura-2-AM. Following preincubation with phillygenin and 1.2 mM CaCl_2_ for 10 min, the synaptosomes were depolarized using 1 mM 4-AP. Using an FL-6500 spectrofluorometer, Fura-2 fluorescence was assessed through dual-wavelength measurements at an emission wavelength of 505 nm and excitation wavelengths of 340 nm and 380 nm. The data were collected at 4-s intervals, and the calcium concentration was subsequently determined [28].

### 2.7. Western Blotting

Synaptosomes were harvested and lysed in ice-chilled RIPA lysis buffer containing protease and phosphatase inhibitors for 10 min. The protein concentration was determined using a Pierce BCA protein assay kit. After separating an equivalent volume of synaptosomal proteins by SDS–PAGE, the proteins were transferred onto a PVDF membrane (Bio-Rad Laboratories, Inc., Hercules, CA, USA). The membranes were blocked with Tris-buffered saline (TBS-T, 0.1% Tween 20) containing 3% BSA and incubated with specific primary antibodies (phospho-ERK 1/2 (Thr202/Tyr204) mAb, 1:2000; ERK 1/2 (Thr202/Tyr204) mAb; phospho-synapsin-I (Ser62/Ser67) pAb, 1:1000; synapsin I mAb, 1:1000; and β-actin, 1:1000) overnight at 4 °C. The membrane was then washed and incubated with horseradish peroxidase-conjugated donkey anti-rabbit secondary antibody (1:8000; BioLegend, San Diego, CA, USA) for 1 h at room temperature, followed by visualization via an enhanced chemiluminescence (ECL) reaction (Merck, Darmstadt, Germany). The expression of phosphorylation levels were quantified via densitometry and are expressed as the normalized relative band density. Densitometric quantification was performed using Multi Gauge software (version 3.0; Fujifilm, Tokyo, Japan).

### 2.8. Data Analysis

The data are presented as the mean ± standard error of the means (SEMs). Statistical analyses were performed using two-tailed Student’s *t* tests or one-way repeated measures analysis of variance (ANOVA) for comparisons among three or more distinct groups. Post hoc analysis was conducted using Tukey’s honest significant difference (HSD) test following one-way ANOVA. The analysis was carried out using GraphPad Prism (version 8.4.3). *p* < 0.05 was considered to indicate statistical significance.

## 3. Results

### 3.1. Phillygenin Inhibits Glutamate Release in Rat Cerebrocortical Synaptosomes

The impact of phillygenin (Figure 1A) on glutamate release was investigated using 4-AP, which is a voltage-sensitive K^+^-channel blocker widely used as a depolarizing agent. In the presence of calcium (Ca^2+^), the addition of 1 mM 4-AP resulted in a progressive increase in glutamate release from cerebral synaptosomes (7.61 ± 0.19 nmol mg^−1^ protein per 5 min; Figure 1B, F (6, 294) = 3.824, *p* < 0.0011). When phillygenin (5, 10, 20, or 30 μM) was added before depolarization with 4-AP, the total glutamate release increased by 6.81 ± 0.18 nmol mg^−1^ protein per 5 min, 6.4 ± 0.25 nmol mg^−1^ protein per 5 min, 5.32 ± 0.38 nmol mg^−1^ protein per 5 min, and 4.60 ± 0.29 nmol mg^−1^ protein per 5 min, respectively (Figure 1A). Dose–response curve analysis yielded an IC50 value of 17 μM. (Figure 1C).

### 3.2. Effects of Calcium Ion Chelators, Glutamate Transporter Inhibitors or Vesicular Transporter Inhibitors on the Inhibition of 4-AP-Evoked Glutamate Release by Phillygenin

The combination of Ca^2+^-dependent and Ca^2+^-independent release mechanisms contributes to the overall release of glutamate [30]. We then explored whether the impact of phillygenin on release was indicative of its influence on either Ca^2+^-dependent exocytotic vesicular release or the Ca^2+^-independent release of glutamate, which can be attributed to the reversal of glutamate efflux by the glutamate transporter. Figure 2A shows that 1.2 mM 4-AP stimulated 1.35 ± 0.15 nmol mg^−1^ glutamate release over 5 min in the presence of calcium-free medium containing 300 μM EGTA. Under these conditions, 20 μM of phillygenin did not alter the Ca^2+^-independent release of glutamate induced by 4-AP. These findings suggest that phillygenin selectively targets and modulates the Ca^2+^-dependent, exocytotic component of glutamate release. To validate this hypothesis and examine the effect of phillygenin on glutamate release, bafilomycin A1, a potent inhibitor of cellular autophagy, or dl-TBOA, a competitive, non-transportable blocker of all excitatory amino acid transporter subtypes, was utilized. As shown in Figure 2B, bafilomycin A1, which depletes synaptic vesicles of glutamate by inhibiting the vesicular glutamate transporter, reduced the amount of 1 mM 4-AP-evoked glutamate release to 1.75 ± 0.13 nmol mg^−1^ protein per 5 min (F (3, 168) = 13.14, *p* < 0.0001). However, in the presence of bafilomycin A1, phillygenin-induced inhibition of 4-AP-evoked glutamate release was absent. Moreover, dl-TBOA suppressed glutamate uptake, leading to a significant increase in 4-AP-induced glutamate release (from 7.36 ± 0.29 to 13.49 ± 0.48 nmol mg^−1^ protein per 5 min). Figure 2C shows that the addition of dl-TBOA completely blocked phillygenin inhibition of 4-AP-induced glutamate release (F (3, 168) = 6.733, *p* < 0.0003). Taken together, the suppression of phillygenin-mediated inhibition of 4-AP-induced glutamate release is observed in the presence of the calcium chelator EGTA or the vesicle transporter inhibitor bafilomycin A1, but not affected by the glutamate transporter inhibitor dl-TBOA (Figure 2D, F (7, 32) = 53.01, *p* < 0.0001). These findings indicate that the phillygenin-mediated inhibition of 4-AP evoked glutamate release is attributed to a decrease in the Ca^2+^-dependent exocytosis of glutamate release.

### 3.3. Effect of Phillygenin on Nerve Terminal Excitability and 4-AP Induced Ca^2+^ Influx

To further understand the mechanism underlying phillygenin-mediated inhibition of glutamate release, we observed the synaptosomal plasma membrane potential and monitored Ca^2+^ influx under depolarizing conditions. The synaptosomal plasma membrane potential was assessed with the voltage-sensitive fluorescent probe DiSC3(5). Table 1 shows that 4-AP administration resulted in an increase in DiSC3(5) fluorescence. (25.63 ± 0.86 units per 5 min, Table 1). The application of phillygenin (20 μM) for 10 min prior to the addition of 4-AP had no effect on the resting plasma membrane potential and did not significantly impact the 4-AP-induced increase in DiSC3(5) fluorescence (27.08 ± 1.47 units per 5 min). Furthermore, we validated phillygenin-mediated inhibition of glutamate release using a high external KCl concentration as an alternative secretagogue. Elevated extracellular KCl levels induce depolarization of the plasma membrane, leading to Ca^2+^ influx through VDCCs into the presynaptic terminal and subsequent neurotransmitter release from synaptic vesicles, which is Na^+^ channel independent. The addition of 15 mM KCl resulted in glutamate release at a rate of 5.18 ± 0.4 nmol mg^−1^ protein per 5 min, which decreased to 2.65 ± 0.63 nmol mg^−1^ protein per 5 min in the presence of 20 μM phillygenin (Table 1). Furthermore, the calcium-sensitive fluorescent dye Fura-2-AM was used to determine the effect of phillygenin on [Ca^2+^]_i_. After applying 1 mM 4-AP, [Ca^2+^]_i_ increased from 218.2 ± 10.62 nM to a plateau of 327.48 ± 19.29 nM in the synaptosomes. Synaptosomes pretreated with 20 μM phillygenin maintained resting Ca^2+^ levels at 211.38 ± 8.29 nM but showed a decrease in 4-AP-induced [Ca^2+^]_i_ to 240.16 ± 11.23 (Table 1). These findings suggest that the observed phillygenin-mediated inhibition of glutamate release is due to a direct reduction in Ca^2+^ entry through VDCCs rather than modulation of the plasma membrane potential.

### 3.4. Effects of Ca^2+^-Channel Antagonists, Intracellular Ca^2+^ Release Inhibitors or Mitochondrial Na^+^/Ca^2+^ Exchanger Inhibitors on Phillygenin-Mediated Inhibition of 4-AP-Induced Glutamate Release

The release of glutamate induced by depolarization is attributed to the entry of Ca^2+^ through different types of Ca^2+^ channels in the plasma membrane, as well as the release of Ca^2+^ into the cytoplasm from intracellular storage compartments such as the endoplasmic reticulum (ER) and mitochondria [31,32]. At central excitatory synapses, the release of presynaptic glutamate is primarily controlled by N-type (Ca_v_2.2) and P/Q-type (Ca_v_2.1) calcium channels, which can be blocked by ω-conotoxin GVIA and ω-agatoxin IVA. Subsequently, we evaluated the specific Ca^2+^ source involved in phillygenin-mediated inhibition of 4-AP-induced glutamate release. As shown in Figure 3A,B,E, 4-AP (1 mM)-evoked glutamate release was significantly reduced in the presence of the ω-conotoxin GVIA (1 μM) or ω-agatoxin IVA (0.1 μM) to 4.5 ± 0.28 and 3.82 ± 0.27 nmol mg^−1^ protein per 5 min, respectively. With the addition of ω-agatoxin IVA, phillygenin continued to inhibit glutamate release induced by 4-AP (1.4 ± 0.24 nmol mg^−1^ protein per 5 min (Figure 3B,E), F (3, 164) = 13.17, *p* < 0.0001). There was no significant difference in glutamate release following treatment with ω-conotoxin GVIA alone compared to that after combined treatment with ω-conotoxin GVIA and phillygenin (*p* = 0.97), suggesting that the inhibitory effect of phillygenin on 4-AP-induced glutamate release may be linked to a reduction in Ca^2+^ influx through Ca_v_2.2 calcium channels but not Ca_v_2.1 calcium channels.

Using the ER Ca^2+^ release blocker dantrolene and the mitochondrial Na^+^/Ca2^+^-exchanger inhibitor CGP37157, we demonstrated that the inhibitory effect of phillygenin on glutamate release is not mediated by reductions in Ca^2+^ release from intracellular stores. The administration of dantrolene (10 μM) resulted in a reduction in the level of 4-AP (1 mM)-evoked glutamate release (4.77 ± 0.42 nmol mg^−1^ protein per 5 min (Figure 3C,E), F (3, 160) = 5.737, *p* < 0.0009). However, in the presence of 20 μM phillygenin, a significant decrease in the level of 4-AP-evoked glutamate release was still observed (*p* < 0.05). Similar results were obtained with the use of 10 μM CGP37157, which effectively inhibited Ca^2+^ efflux from the mitochondria (Figure 3D,E), F (3, 164) = 11.29, *p* < 0.0001.

### 3.5. Phillygenin Inhibits 4-AP-Evoked Glutamate Release through Extracellular Signal-Regulated Kinase Signaling

Extracellular signal-regulated kinase (ERK), protein kinase C (PKC), and protein kinase A (PKA) are present at the presynaptic level and play crucial roles in neurotransmitter release [33,34]. In this study, we examined the protein kinase cascade implicated in phillygenin-induced suppression of 4-AP-evoked glutamate release. Figure 4A shows that at a concentration of 20 μM, the ERK pathway inhibitor PD98059 effectively reversed the inhibitory effect of phillygenin on 4-AP-induced glutamate release (*p* = 0.3, Figure 4A,E). A similar result was obtained when synaptosomes were subjected to treatment with FR180204 (10 μM), which is a potent, selective, cell-permeable inhibitor of ERK1 and ERK2 (*p* = 0.76, Figure 4B,E). In contrast, the PKA inhibitor H89 (10 μM) (Figure 4C,E, F (3, 168) = 4.800, *p* < 0.0031) and the PKC inhibitor GF109203X (5 μM) (Figure 4D,E, F (3, 168) = 6.793, *p* < 0.0002), which individually suppressed 4-AP-induced glutamate release, did not have any observable effect on the phillygenin-mediated inhibition of 4-AP-evoked glutamate release. These findings suggest that the inhibition of glutamate release by phillygenin is associated with the MAPK/ERK signaling pathway.

To further confirm the role of the MAPK/ERK signaling cascade in phillygenin-mediated inhibition of glutamate release, western blotting analysis was used to assess the phosphorylation of ERK1/2. Figure 5 shows that the depolarization of purified synaptosomes with 1 mM 4-AP led to a notable increase in ERK1/2 phosphorylation. Importantly, this effect was effectively inhibited by phillygenin. As a substrate for ERK protein kinases, synapsin I is a phosphoprotein associated with vesicles and localized at presynaptic terminals. The crucial function of synapsin I involves the regulation of vesicle dynamics and the release of neurotransmitters. Similar results were obtained from the analysis of synapsin I phosphorylation. The presence of 1 mM 4-AP led to an increase in the phosphorylation of synapsin I (135.91% ± 6.07%; *p* < 0.01), and this effect was diminished following treatment with phillygenin (89.53% ± 16.02%; *p* < 0.01).

## 4. Discussion

In this study, we present a novel observation using nerve terminals isolated from the cerebral cortex of a rat model to demonstrate the inhibitory effect of phillygenin on 4-AP-induced glutamate release. The ability of phillygenin to inhibit 4-AP-induced glutamate release suggests it can be used to limit excessive glutamate release, a major pathogenetic mechanism in several neurological disease states, including ischemic brain damage and neurodegeneration [35,36].

The release of neurotransmitters at synapses is a complex process linked to membrane depolarization and involves the regulation of ion channels, including Na^+^, K^+^, and Ca^2+^ channels [37,38]. Inhibiting Na^+^ channels and activating K^+^ channels shortened the duration of action potentials and stabilized membrane excitability. This ultimately causes a subsequent reduction in Ca^2+^ entry and the release of neurotransmitters [39,40]. Therefore, the observed reduction in glutamate release suggested that phillygenin activated a significant protective mechanism in response to excitotoxic insults. Our objective was to investigate the underlying mechanisms responsible for phillygenin-mediated inhibition of glutamate release, focusing on the intrasynaptosomal Ca^2+^ concentration, the synaptosomal plasma membrane potential, VDCCs, and the activation of protein kinases in rat brain synaptosomes. The following discussion outlines potential mechanisms involved in phillygenin-mediated inhibition of glutamate release.

In this study, we examined phillygenin-mediated inhibition of glutamate release by exploring two potential mechanisms: first, the modification of the synaptosomal plasma membrane potential and second, the direct regulation of Ca^2+^ entry through VDCCs. The first scenario seems unlikely for three reasons. (1) Under both resting conditions and during depolarization with 4-AP, phillygenin did not significantly impact the synaptosomal plasma membrane potential, suggesting limited influence on K^+^ conductance. (2) Phillygenin had no effect on the Ca^2+^-independent release of glutamate triggered by 4-AP, a process that is solely dependent on the membrane potential [41]. In accordance with these findings, the inhibitory effect of PD on 4-AP-induced glutamate release was prevented by the cellular autophagy inhibitor, bafilomycin A1, but remained unaffected by the presence of dl-TBOA, an inhibitor of excitatory amino acid transporters (EAATs). This observation suggested that phillygenin does not influence the release of glutamate by altering the direction of the plasma membrane glutamate transporter. (3) The release of glutamate triggered by 4-AP is associated with Na^+^ and Ca^2+^ channels, while KCl-induced release involves only Ca^2+^ channels [42]. Phillygenin significantly inhibited both 4-AP- and KCl-evoked glutamate release, suggesting that the effect of phillygenin involves Ca^2+^ channels rather than Na^+^ channels. These findings strongly indicate that phillygenin-mediated inhibition of 4-AP-evoked glutamate release is associated with a reduction in Ca^2+^-dependent exocytosis.

At synaptic terminals, cytoplasmic Ca^2+^ results from the combined contribution of extracellular Ca^2+^ influx through plasma membrane VDCCs and intracellular Ca^2+^ release [31,43]. An increase in cytoplasmic Ca^2+^ levels is associated with the release of glutamate. Using the endoplasmic reticulum (ER) ryanodine receptor inhibitor, dantrolene, and the mitochondrial Na^+^/Ca^2+^ exchange inhibitor, CGP37157, we demonstrated that phillygenin has no effect on intracellular Ca^2+^ release. These data suggest that intracellular Ca^2+^ stored in the ER and mitochondria was not involved in phillygenin inhibition of 4-AP-evoked glutamate release. Phillygenin suppressed the 4-AP-induced increase in [Ca^2+^]_i_ when the calcium indicator Fura-2-AM was utilized. These findings suggested that phillygenin inhibits glutamate release by reducing presynaptic Ca^2+^ influx through VDCCs. Moreover, the inhibitory effect of phillygenin on 4-AP-induced glutamate release from synaptosomes was abolished in the presence of the Ca_v_2.2 (N-type) calcium channel blocker ω-conotoxin GVIA. In contrast, the action of phillygenin was not affected when the Ca_v_2.1 (P/Q-type) calcium channel was blocked by ω-agatoxin IVA. These findings indicate that the modulation of glutamate release by phillygenin is linked to the inhibition of Ca^2+^ influx through presynaptic Ca_v_2.2 calcium channels. The mechanism by which phillygenin modulates Ca_v_2.2 calcium channels remains unclear. Phillygenin’s effect may occur directly on presynaptic Ca_v_2.2 channels or through another way such as the modulation of protein kinase activity, altering VDCCs phosphorylation. Therefore, additional research is needed to fully understand phillygenin’s impact on Ca_v_2.2 calcium channels.

The effect of phillygenin on Ca_v_2.2 calcium channels may be attributed to indirect modulation through a series of protein interactions between the membranes of synaptic vesicles and presynaptic terminals [44,45]. The activity of presynaptic VDCCs and glutamate release are known to be regulated by second messenger-activated protein kinases, including PKA, PKC, and MAPK/ERK [46,47]. In this investigation, the effect of phillygenin on 4-AP-induced glutamate release was effectively blocked by the ERK inhibitors, PD98059 and FR180204. Conversely, the decrease in 4-AP-induced glutamate release caused by phillygenin remained unaffected when synaptosomes were subjected to incubation with the PKA inhibitor, H89 and the PKC inhibitor, GF109203X. These results suggest that phillygenin inhibits glutamate release from rat cerebral cortex nerve terminals through the MAPK/ERK kinase cascade. Furthermore, phillygenin significantly reduced the phosphorylation of ERK1/2 and synapsin I at the ERK1/2-dependent phosphorylation sites 4/5 induced by 4-AP. The MAPK/ERK cascades are key signaling pathways that regulate neurotransmitter exocytosis by transmitting extracellular signals to intracellular targets. Ca^2+^ influx, induced by depolarization, activates MAPK/ERK and phosphorylates synapsin I at sites 4/5. This phosphorylation of synapsin I releases synaptic vesicles from the actin cytoskeleton in response to specific stimuli. As a result, more vesicles are available near the active zone for neurotransmitter exocytosis, leading to the release of glutamate from a storage vesicle into the synaptic cleft [48]. These findings suggest that phillygenin inhibits glutamate release by suppressing MAPK/ERK-dependent synapsin I phosphorylation and reducing the number of synaptic vesicles.

Elevated levels of extracellular glutamate in the brain can lead to neuronal damage via excitotoxicity, which is a critical process in neurodegeneration. Compounds capable of modulating the release of glutamate show promise as therapeutic agents for neuroprotection. Phillygenin, a lignan compound from a medicinal herb, has shown promising therapeutic potential due to its effects on inflammation, oxidation, tumors, bacteria, viruses, pain, and liver damage [13,16,20,49]. Oxidative stress can trigger free radical attack of neuronal cells, which contributes to the development of neurodegenerative disorders. Phillygenin, a natural antioxidant, has the potential to protect the brain against the damaging effects of free radicals and inflammation and potentially offer neuroprotective benefits. Most experiments have demonstrated phillygenin’s non-toxicity to cells or experimental animals. Additionally, acute toxicity tests in mice showed no adverse effects even at high doses, indicating its safety [50,51]. In the present study, phillygenin inhibited the Ca^2+^-dependent exocytosis of glutamate in a dose-dependent manner in a concentration range of 5–50 μM and an IC50 value of 17 μM. Lin et al. demonstrated that phillygenin at a dose of 30–100 μM was effective in suppressing the inflammatory response and inhibiting apoptosis in vitro [23]. In addition, in vitro studies have shown that phillygenin exerts antitumor effects at concentrations ranging from 10–100 μM [14,18,52]. The results of the present study are consistent with these reports; however, the antioxidant effects of phillygenin were observed at relatively high IC50 values (approximately 140 μM) [12,53].

While the precise mechanisms underlying the neuroprotective effect of phillygenin are yet to be fully understood, there have been reports suggesting the potential involvement of scavenging free radicals or exhibiting antioxidant properties [16,53]. In this study, the ability of phillygenin to reduce glutamate release from nerve terminals may partially elucidate its neuroprotective mechanism. In addition, phillygenin exhibits significant analgesic activities and may interact with glutamatergic receptors or signaling pathways to attenuate pain perception [49]. Excessive glutamate release and excitotoxicity are linked to pain conditions and neuropathic pain. Phillygenin’s ability to regulate glutamate levels or transmission may contribute to its analgesic effects and offer neuroprotection against glutamate-induced neuronal damage. However, the limited research on phillygenin’s neuroprotection necessitates further investigation in future studies.

In summary, the findings from this study suggest that phillygenin reduces glutamate release from rat brain nerve endings by blocking presynaptic Ca^2+^ entry through Ca_v_2.2 calcium channels. These findings highlight a crucial mechanism by which phillygenin may protect neurons against excitotoxic damage induced by calcium overload. Moreover, the observed suppression may, at least partially, be influenced by inhibition of the MAPK/ERK/synapsin I pathway (Figure 6).

## 5. Conclusions

Phillygenin decreases glutamate release by inhibiting Ca_v_2.2 (N-type) calcium channels and thus affecting the MAPK/ERK/synapsin I pathway in rat cerebrocortical synaptosomes. Our finding is crucial for understanding the role of phillygenin in the brain and for exploiting its potential in therapeutic interventions.

## Figures and Tables

**Figure 1 biomedicines-12-00495-f001:**
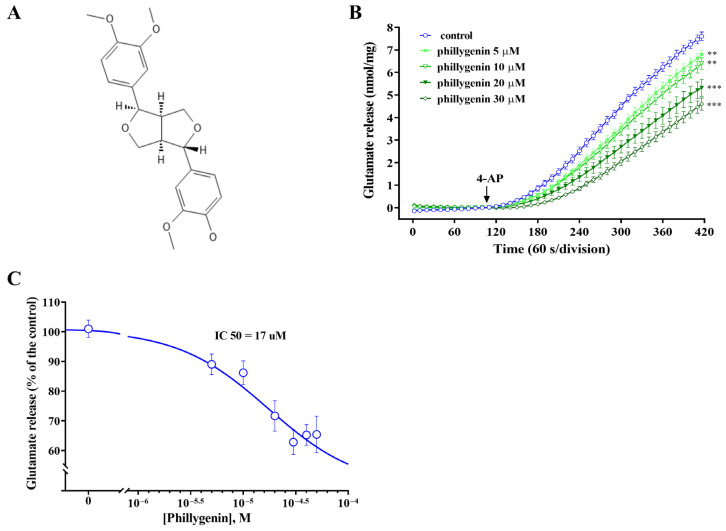
Phillygenin suppresses the release of glutamate induced by 4-AP in a concentration-dependent manner. Rat synaptosomes were resuspended in HBM buffer at a final protein concentration of 0.5 mg/mL and incubated for 3 min followed by the addition of 1 mM CaCl_2_. After an additional 10 min, 1 mM 4-AP was introduced to induce depolarization (indicated by an arrow). The release of glutamate was assessed using a continuous fluorometric assay. (**A**) Chemical structure of phillygenin. (**B**) Glutamate release was evaluated through a continuous fluorometric assay, in either control conditions and in the presence of 5–30 μM phillygenin, administered 10 min prior to the introduction of 4-AP. (**C**) A concentration-dependent reduction was observed in 4-AP-stimulated glutamate release in the presence of phillygenin. The results represent the mean ± standard error of the mean (S.E.M.) values from independent experiments utilizing synaptosomal preparations from six animals. Mean and S.E.M. were calculated at 2-s intervals, with error bars depicted at intervals of 10 s for better clarity. ** *p* < 0.01 compared with the control group, *** *p* < 0.001 compared with the control group.

**Figure 2 biomedicines-12-00495-f002:**
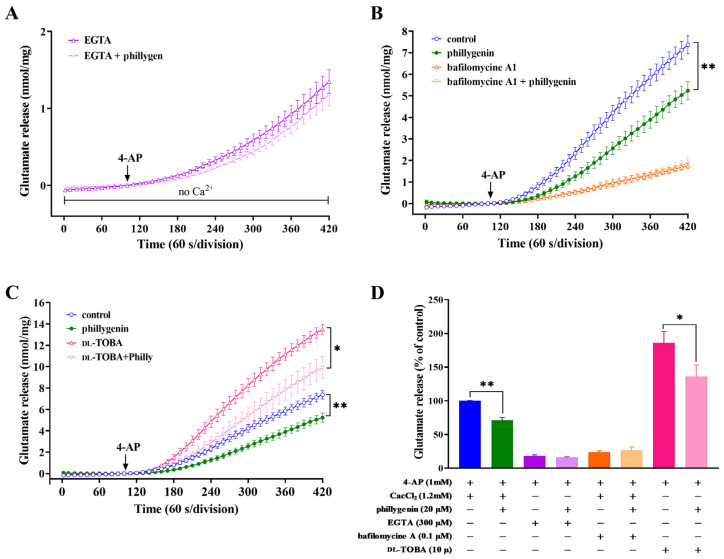
The effect of external calcium omission, the glutamate transporter blocker dl-TBOA, and the vesicular transporter inhibitor bafilomycin A1 on the inhibition of 4-AP-evoked glutamate release mediated by phillygenin. (**A**) Ca^2+^-independent release was assessed by excluding CaCl_2_ and introducing 300 μM EGTA 10 min before depolarization. The release was triggered by 1 mM 4-AP, both in control conditions and in the presence of 20 μM phillygenin, administered 10 min prior to the introduction of 4-AP. The effect of phillygenin on 4-AP-evoked glutamate release was assessed in the absence (control) and presence of dl-TBOA (10 μM). The black arrow indicates the moment when 4-AP is added. (**B**) and bafilomycin A1 (0.1 μM) (**C**). dl-TBOA, bafilomycin A1 and phillygenin were added 10 min before depolarization. (**D**) Quantitative analysis comparing the amount of glutamate release induced by 1 mM 4-AP in the presence and absence of EGTA, as well as in the presence and absence of dl-TBOA or bafilomycin A1. Results are mean ± S.E.M. of five independent experiments. ** *p* < 0.01 versus the control group, * *p* < 0.05 versus the dl-TBOA-treated group.

**Figure 3 biomedicines-12-00495-f003:**
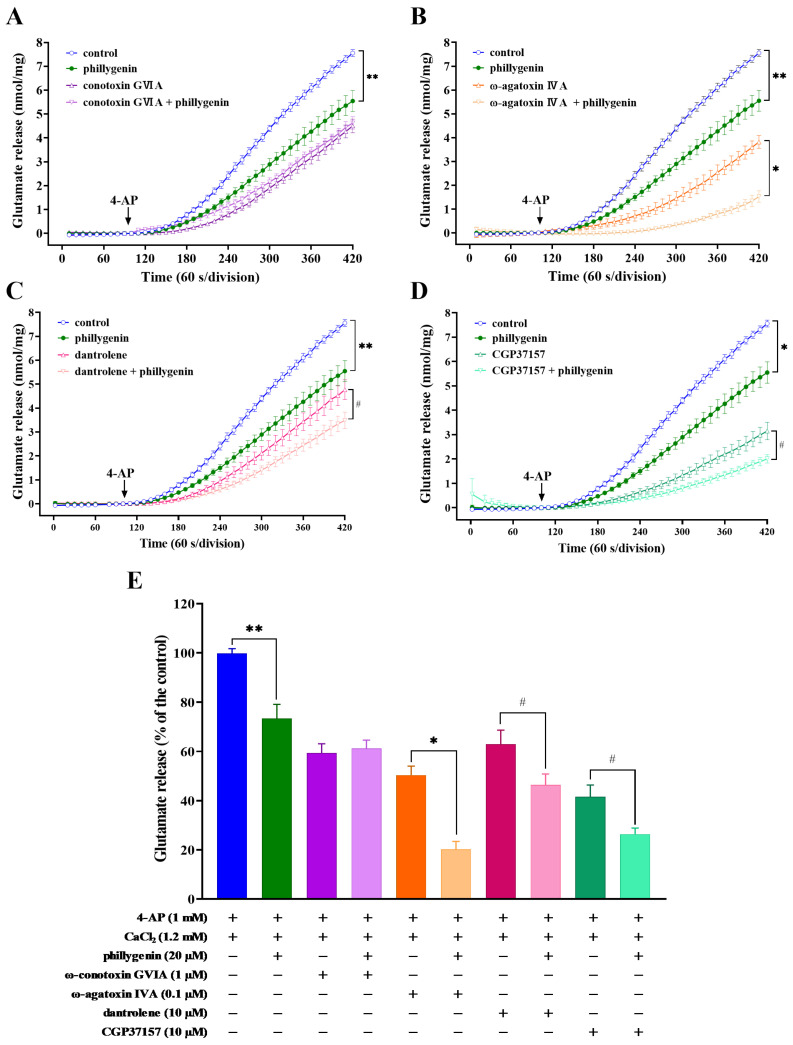
Phillygenin inhibited 4-AP-induced glutamate release by modulating Ca_v_2.2 calcium channels, but not Ca_v_2.1 channels. Glutamate release was induced by 1 mM 4-AP in the absence (control) or presence of 1 μM ω-conotoxin GVIA (**A**), 0.1 μM ω-agatoxin IVA (**B**), 10 μM dantrolene (**C**), or 10 μM CGP37157 (**D**), administered 10 min before the addition of 20 μM phillygenin. (**E**) Quantitative evaluation of the released glutamate levels under different conditions. Results are the mean ± S.E.M. of five independent experiments. ** *p* < 0.01 versus the control group, * *p* < 0.001 versus the ω-agatoxin IVA-treated group, # *p* < 0.05 versus the dantrolene-, or CGP37157-treated group.

**Figure 4 biomedicines-12-00495-f004:**
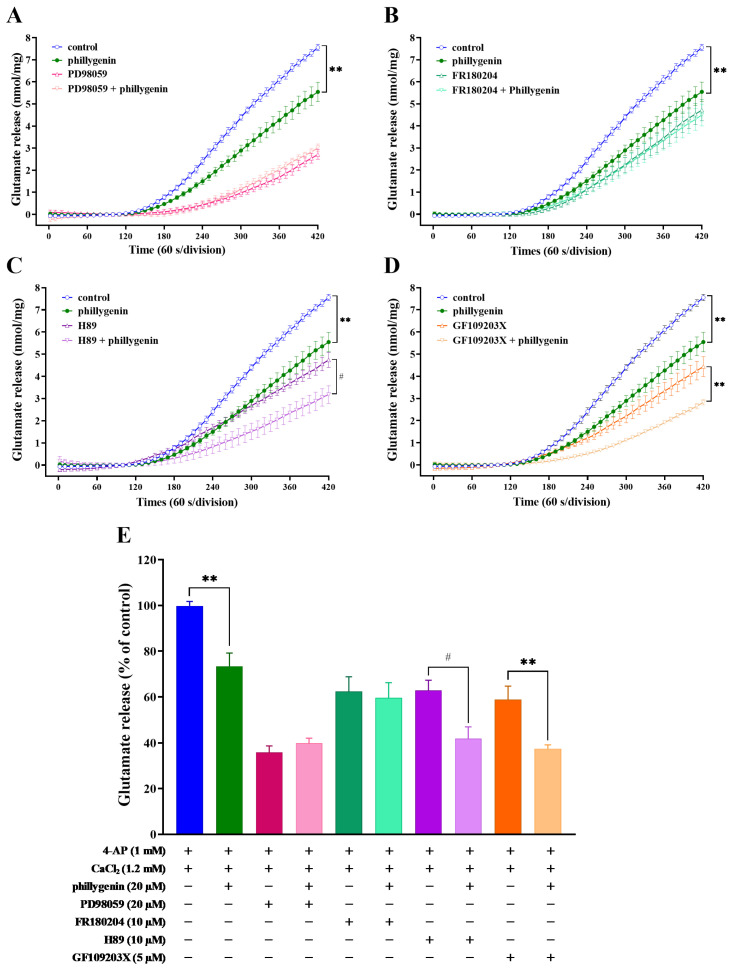
The inhibition of 4-AP-induced glutamate release by phillygenin is entirely blocked in the presence of the ERK inhibitors PD98059 and FR180204. Glutamate release was induced by 1 mM 4-AP in the absence (control) or presence of 20 μM PD98059 (**A**), 10 μM FR180204 (**B**), 10 μM H89 (**C**), or 5 μM GF109203X (**D**), administered 10 min before the addition of 20 μM phillygenin. (**E**) Quantitative evaluation of the released glutamate levels under different conditions. Results are the mean ± S.E.M. of five independent experiments. ** *p* < 0.01 versus the control group, or GF109203X-treated group. # *p* < 0.05 versus the H89-treated group.

**Figure 5 biomedicines-12-00495-f005:**
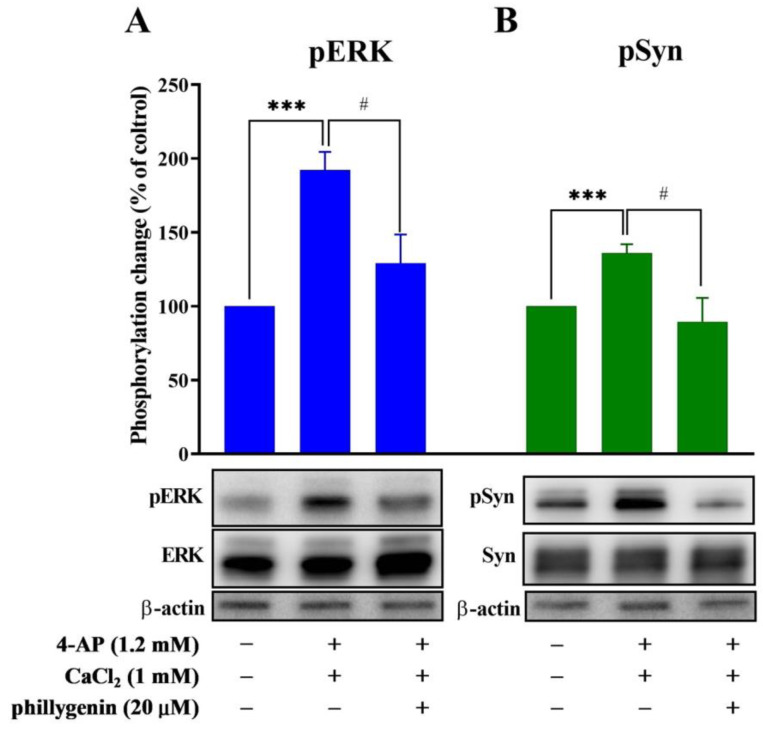
The Effect of phillygenin on the 4-AP-evoked phosphorylation of ERK1/2 and synapsin I, which is a substrate of ERK. Phillygenin was added 10 min before depolarization with 4-AP. The phosphorylation levels of ERK1/2 (**A**) and synapsin I (**B**) in synaptosomes were assessed and expressed as a percentage relative to the measurements obtained from the control group without 4-AP. Each bar represents the means ± S.E.M. of the results obtained in 3 experiments (*n* = 3 per group). *** *p* < 0.001 versus the control group. # *p* < 0.05 versus the 4-AP-treated group.

**Figure 6 biomedicines-12-00495-f006:**
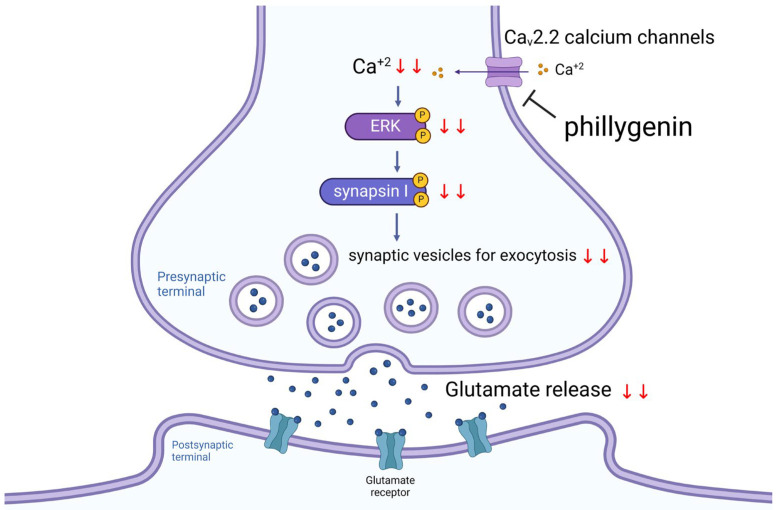
Schematic representation of the main mechanism involved in phillygenin-mediated inhibition of glutamate release from cerebral synaptosomes. Phillygenin suppresses Ca_v_2.2 calcium channels, which in turn inhibits the MAPK/ERK/synapsin I pathway, thus decreasing the amount of glutamate release. Red downward arrows indicate a decrease. Graph created with BioRender.com.

**Table 1 biomedicines-12-00495-t001:** Phillygenin attenuates the 4-AP-induced elevation in [Ca^2+^]_i_ but does not affect the synaptosomal membrane potential. The synaptosomal membrane potential was monitored using DiSC3(5) (5 μM), both in the absence (control) and in the presence of 20 μM phillygenin added 10 min prior to depolarization induced by 1 mM 4-AP. The effect of phillygenin on 15 mM KCl-induced glutamate release was examined. The experiments were conducted as previously described, with the exception that 15 mM KCl was used as a secretagogue instead of 4-AP. The intrasynaptosomal level of Ca^2+^ (nM) was observed using Fura-2 (5 μM), both in the absence (control) and in the presence of 20 μM phillygenin added 10 min before depolarization with 1 mM 4-AP. ** *p* < 0.01 versus the KCl control group, * *p* < 0.001 versus the 4-AP control group.

	Membrane Potential (Fluorescence Units)			Glutamate Release (nmol mg^−1^ Protein per 5 min)			[Ca^2+^]_i_ (nM)		
	Basal	4-AP	*n*	Basal	KCl	*n*	Basal	4-AP	*n*
Control	0.01 ± 0.03	25.63 ± 0.86	5	−0.08 ± 0.08	5.18 ± 0.4	5	218.2 ± 10.62	327.48 ± 19.29	5
Phillygenin	0.02 ± 0.12	27.08 ± 1.47	5	−0.07 ± 0.04	2.65 ± 0.63 **	5	211.38 ± 8.29	240.16 ± 11.23 *	5

Results are expressed as the mean ± S.E.M. of five independent experiments. * *p* < 0.01 versus control, ** *p* < 0.001 versus control.

## Data Availability

Data will be made available on request.

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
