# Peer review of "Phillygenin Suppresses Glutamate Exocytosis in Rat Cerebrocortical Nerve Terminals (Synaptosomes) through the Inhibition of Cav2.2 Calcium Channels"

_biomedicines, 2024, doi:10.3390/biomedicines12030495_

Round 1
Reviewer 1 Report
Comments and Suggestions for Authors
The study conducted by Lee et al. provides valuable insights into the mechanism by which phillygenin mediates the inhibition of evoked glutamate release from synaptosomes. Their findings suggest that phillygenin primarily reduces the influx of Ca2+ through Cav2.2 calcium channels, subsequently suppressing the 38 MAPK/ERK/synapsin I signaling cascade.
While Lee et al.'s study offers significant contributions to our understanding of phillygenin's effects on glutamate release and calcium channel activity, it is natural to question whether phillygenin also impacts the expression of calcium channels themselves.
Exploring the potential effects of phillygenin on calcium channel expression could shed further light on its broader mechanisms of action and provide additional insights into its therapeutic potential. Such investigations could involve assessing changes in calcium channel gene expression levels or protein levels in response to phillygenin treatment using techniques like RT-qPCR or Western blotting.
Comments on the Quality of English Languageminor editing of English language required
Author Response
Response to reviewers
biomedicines-2873173
We thank the reviewer for the critical comments and constructive suggestions.
Response to reviewer1
Comments and Suggestions for Authors
The study conducted by Lee et al. provides valuable insights into the mechanism by which phillygenin mediates the inhibition of evoked glutamate release from synaptosomes. Their findings suggest that phillygenin primarily reduces the influx of Ca2+ through Cav2.2 calcium channels, subsequently suppressing the 38 MAPK/ERK/synapsin I signaling cascade.
While Lee et al.'s study offers significant contributions to our understanding of phillygenin's effects on glutamate release and calcium channel activity, it is natural to question whether phillygenin also impacts the expression of calcium channels themselves.
Exploring the potential effects of phillygenin on calcium channel expression could shed further light on its broader mechanisms of action and provide additional insights into its therapeutic potential. Such investigations could involve assessing changes in calcium channel gene expression levels or protein levels in response to phillygenin treatment using techniques like RT-qPCR or Western blotting.
We agree this point mentioned by the reviewer. In response, we have incorporated a few additional sentences into the Discussion section.
“The mechanism by which phillygenin modulates Cav2.2 calcium channels remains unclear. Phillygenin's effect may occur directly on presynaptic Cav2.2 channels or through other way like modulation of protein kinase activity, altering VDCCs phosphorylation. Therefore, additional research is needed to fully understand phillygenin's impact on Cav2.2 calcium channels.” (Page 16, line 426-430).
The manuscript is edited by the AJE.

Reviewer 2 Report
Comments and Suggestions for Authors
Authors investigated the effect and mechanism of phillygenin, a natural compound with neuroprotective potential, on glutamate release in isolated nerve terminals (synaptosomes) prepared from the rat cerebral cortex. They revealed that phillygenin reduced 4-AP-induced glutamate release. The effect of phillygenin on glutamate release triggered by 4-AP was completely abolished when MAPK/ERK inhibitors were applied. They also showed that phillygenin attenuated the phosphorylation of ERK1/2 and its major presynaptic target, synapsin I, a protein associated with synaptic vesicles.
It is a well-presented work and it can be accepted after minor revision.
Here are some points:
1. Authors shoud add the chemical structure of phillygenin as a Figure.
2.Can phillygenin be adequately accumulated into nerve terminals to show its pharmacological effects? Authors can calculate some pharmacokinetic parameters with some in silico techniques or at least they can mention them in Introduction.
2. Authors can explain the effects of phillygenin against neurodegeneration in detail in Introduction.
3. In Discussion, authors can mention that in which disorders phillygenin can be beneficial.
4. Authors can compare these findings with similar studies in Discussion.
5. Authors can add Conclusion part, too for brief presentation of their findings.
6. What about the toxicity of phillygenin? Is it safe to use or are there any other studies related to this topic?
Author Response
Response to reviewers
biomedicines-2873173
We thank the reviewer for the critical comments and constructive suggestions.
Response to reviewer2
- Authors shoud add the chemical structure of phillygenin as a Figure.
As suggested by the reviewer, the chemical structure of phillygenin has been included in Figure 1A. (Page 5, Line 199-Page 6, Line 215)
- Can phillygenin be adequately accumulated into nerve terminals to show its pharmacological effects? Authors can calculate some pharmacokinetic parameters with some in silico techniques or at least they can mention them in Introduction.
As suggestion by the reviewer, serveral sentences have been added. “The pharmacokinetics of phillygenin in rats exhibit first-order kinetics, with rapid distribution and elimination, while also showing high oral bioavailability in mice, peaking within 30 minutes.” (Page 2, Line 63-65)
- Authors can explain the effects of phillygenin against neurodegeneration in detail in Introduction.
The sentence is modified to enhance clarity. “Moreover, phillygenin possesses anti-inflammatory properties, which may provide neuroprotective benefits by potentially reversing cellular damage and slowing the progression of neuronal cell loss in individuals with neurodegenerative disorders.” (Page 2, Line 71-73)
- In Discussion, authors can mention that in which disorders phillygenin can be beneficial.
As suggestion by the reviewer, serveral sentences have been added. “The ability of phillygenin to inhibit 4-AP-induced glutamate release suggests it can be used to limit excessive glutamate release, a major pathogenetic mechanism in several neurological disease states, including ischemic brain damage and neurodegeneration [35,36].” (Page 15, Line 375-379)
- Authors can compare these findings with similar studies in Discussion.
We thank the reviewer for the suggestion. In response, we have incorporated a few additional sentences into the Discussion section.
“While the precise mechanisms underlying the neuroprotective effect of phillygenin are yet to be fully understood, there have been reports suggesting the potential in-volvement of scavenging free radicals or exhibiting antioxidant properties [16,53]. In this study, the ability of phillygenin to reduce glutamate release from nerve terminals may partially elucidate its neuroprotective mechanism. In addition, phillygenin exhibits significant analgesic activities and may interact with glutamatergic receptors or signaling pathways to attenuate pain perception [49]. Excessive glutamate release and excitotoxicity are linked to pain conditions and neuropathic pain. Phillygenin's ability to regulate glutamate levels or transmission may contribute to its analgesic effects and offer neuroprotection against glutamate-induced neuronal damage. However, the limited research on phillygenin's neuroprotection necessitates further investigation in future studies.” (Page 16, Line 471- Page 17, Line 481)
- Authors can add Conclusion part, too for brief presentation of their findings.
We thank the reviewer for the suggestion. The Conclusion has been added. “Phillygenin decreases glutamate release by inhibiting Cav2.2 (N-type) calcium channels and MAPK/ERK/synapsin I pathway in rats cerebrocortical synaptosomes. Our finding is crucial for understanding the role of phillygenin in the brain and for exploiting its potential in therapeutic interventions.” (Page 18, Line 492-496)
- What about the toxicity of phillygenin? Is it safe to use or are there any other studies related to this topic?
As suggestion by the reviewer, serveral sentences have been added. “Most experiments have demonstrated phillygenin's non-toxicity to cells or experimental animals. Additionally, acute toxicity tests in mice showed no adverse effects even at high doses, indicating its safty [50,51].” (Page 17, Line 461-463)
The manuscript is edited by the AJE.
